



**Effects of Peatland Management on Aquatic Carbon Concentrations and Fluxes**
Amy E. Pickard[1*] Marcella Branagan[1,2] Mike F. Billett[1,3] Roxane Andersen[2], and Kerry J. Dinsmore[1]
[1]UK Centre for Ecology & Hydrology, Edinburgh, Bush Estate, Penicuik, Midlothian, EH26 0QB, UK
*Corresponding author: amypic92@ceh.ac.uk
[2] Environmental Research Institute, University of Highlands and Islands, Castle St., Thurso, KW14 7JD, UK.
[3]Department of Biological and Environmental Sciences, University of Stirling, UK.



## Abstract

Direct land to atmosphere carbon exchange has been the primary focus in previous studies of peatland disturbance and subsequent restoration. However, loss of carbon via the fluvial pathway is a significant term in peatland carbon budgets and requires consideration to assess the overall impact of restoration measures. This study aimed to determine the effect of peatland land management regime on aquatic carbon concentrations and fluxes in an area within the UK's largest tract of blanket bog, the Flow Country of N. Scotland. Three sub catchments were selected to represent peatland land management types: non-drained, drained and restoration (achieved through drain blocking and tree-removal). Water samples were collected on a fortnightly basis from September 2008 to August 2010 at six sampling sites, one located upstream and one downstream within each sub catchment. Concentrations of DOC were significantly lower for the upstream non-drained sub catchment compared to the drained sub catchments, and there was considerable variation in the speciation of aquatic carbon (DOC, DIC, POC, $CO_2$ and $CH_4$) across the monitoring sites, with significantly higher POC concentrations observed in the restored sub-catchment most affected by tree-removal. Aquatic carbon fluxes were highest from the drained catchments and lowest from the non-drained catchments at 25.6 and 10.4 g C $m^{-2}$ $yr^{-1}$, respectively, with variability between the upstream and downstream sites within each catchment very low. It is clear from both the aquatic carbon concentration and flux data that drainage has had a profound impact on the hydrological and biogeochemical functioning of the peatland. In the restoration catchment, carbon export varied considerably, from 23.3 g C $m^{-2}$ $yr^{-1}$ at the upper site to 11.4 g C $m^{-2}$ $yr^{-1}$ at the lower site, largely due to differences in runoff generation. As a result of this hydrological variability it is difficult to make definitive conclusions about the impact of restoration on carbon fluxes and further monitoring is needed to corroborate the longer term effects.

## Keywords

Flow Country, Aquatic Carbon Fluxes, DOC, Peatlands, Drainage, Ditch Blocking



## 1. Introduction

The ability of peatlands to store and sequester carbon is of major importance both nationally in terms of greenhouse gas (GHG) accounting, and globally in understanding the carbon cycle and potential changes to atmospheric composition. Loss of carbon via the aquatic pathway constitutes a significant term within peatland carbon budgets, in some past studies accounting for between 34% and 51% of uptake from net ecosystem exchange (NEE) (Dinsmore et al., 2010; Nilsson et al., 2008; Roulet et al., 2007). Aquatic carbon fluxes include dissolved and particulate organic carbon (DOC and POC), dissolved inorganic carbon (DIC), and within this, gaseous carbon in the form of carbon dioxide ($CO_2$) and methane ($CH_4$). Fluvial export of DOC is typically the largest aquatic flux, with losses from UK peatland catchments in the range 19 to 27 g C m$^{-2}$ yr$^{-1}$ (Billett et al., 2010). Accordingly, DOC is also the most frequently reported of the aquatic carbon fluxes.

Whilst there is considerable inter-annual variability evident in many of the carbon flux pathways from peatlands (e.g. Dinsmore et al., 2013; Helfter et al., 2015), a significant increasing trend in DOC concentrations has been detected in the majority of monitored surface waters in Europe and North America since the 1980s (Monteith et al., 2007). On the regional scale this trend has largely been attributed to recovery of soils from acid deposition (Evans et al., 2012; Monteith et al., 2007), however on the catchment scale, anthropogenic disturbance of peatlands has been identified as a potential contributing factor to the observed DOC increases (Billett et al., 2010; Parry et al., 2014).

Anthropogenic disturbance covers a range of activities including burning, peat cutting and afforestation, with peatland drainage by far the most prevalent form of disturbance. It is estimated that 447,637 km$^2$ of peatlands are drained globally, releasing up to 1,058 Mt $CO_2$ annually (Joosten, 2010), with a shift in the global peatland biome from a net sink to a net source of C thought to have occurred in the 1960s (Leifeld et al., 2019). The UK alone is thought to produce approximately 9.6 Mt $CO_2$ yr$^{-1}$ from degraded, often drained peatlands (Bain et al., 2011). Drainage results in erosion and a lowering of the water table, which exposes greater peat depths to aerobic conditions. Although the exact response differs between peatland types and with time since disturbance (Laiho, 2006), artificially lowering the water table is generally understood to increase decomposition rates. This results in a larger pool of soluble carbon species that can be transported via soil throughflow to the surface drainage system, where increases in DOC concentrations are subsequently detected (Evans et al., 2016; Menberu et al., 2017; Strack et al., 2008; Worrall et al., 2004). Notably in Great Britain,



upland conifer plantations including those on drained, deep peat are estimated to have raised the overall DOC
export by as much as 0.168 Tg C year$^{-1}$ (Williamson et al., 2021).
In recognition of the value of intact peatlands there is now a significant national and international effort to
reduce peatland drainage and focus on restoration activities (Parry et al., 2014). In most cases the primary goal
of restoration is to return the hydrological functioning of the peatland to the assumed pre-management state as
a precursor for re-establishing the lost ecosystem functioning. Drain blocks are a cost-effective means by which
to raise the water table of human-impacted peatlands and are constructed using a variety of damming methods
such as plastic piling, heather bales or peat dams (Armstrong et al., 2009; Parry et al., 2014). Their
implementation in previously drained catchments has in many cases resulted in successful re-wetting of
peatlands (Strack and Zuback, 2013; Waddington and Price, 2000) and reductions in peak discharge
(Shuttleworth et al., 2019). However the degree of their success has been shown to be spatially variable as a
function of ditch direction across the slope and height of water table prior to intervention (Holden et al., 2017a).
Associated reductions in DOC concentrations and fluxes are often an assumed co-benefit of restoration via
drain blocking and, therefore, this practice has been funded by water companies that source water from peat
catchments in an effort to reduce DOC concentrations in their pre-treatment raw water (Andersen et al., 2017).
Despite this assumed co-benefit, the reported effects of drain blocking on concentrations of DOC are not
consistent and often show contradictory results depending on time since blocking. Increases in concentrations
have been seen up to two years after restoration (Gibson et al., 2009; Worrall et al., 2007), while studies
conducted three to four years after blocking report lower concentrations in soil and stream water (Wallage et
al., 2006; Wilson et al., 2011). In a paired catchment study with an extended baseline data collection period
(three years pre-blocking), drain blocking showed no discernible impact on DOC or other measured carbon
species in ditch waters and stream waters after six years (Evans et al., 2018). The balance of evidence suggests
that different peatlands will display variable water quality responses to drain blocking controlled by factors
such as slope, altitude, rainfall, and further research is required to understand what drives different response
mechanisms.
Determining the effect of drain-blocking can be further complicated or masked by other simultaneous
restoration works, for example, removal of trees from peat with heavy machinery, which has previously been





88 shown to result in short-term increases in aquatic DOC concentrations (Zheng et al., 2018; Gaffney et al.,

89 2020). The blanket bogs of the Flow Country have been subject to multiple and changing land management

90 practices over the past half century. Afforestation of the Flow Country peatlands occurred during the 1970s

91 and 1980s and areas designated for planting were first drained to lower the water table and then planted with

92 non-native conifers (Lindsay et al., 1988). Large-scale "forest-to-bog" restoration, whereby non-native

93 conifers are extracted, drains are blocked and further management (e.g. brash crushing, shredding, peat-

94 reprofiling, etc.), has been on-going since the 1990s in an effort to restore the bog's ecosystem functioning

95 (Andersen et al., 2017). This has resulted in a patchwork of land-use over a relatively small spatial scale, and

96 a unique opportunity to carry out detailed management effects research on quasi replicated catchments that fall

97 within the most extensive area of continuous blanket peatland in Europe (Lindsay et al., 1988), which serves

98 as a nationally important carbon store .

99 Here we utilise the land-use mosaic the Flow Country provides, monitoring aquatic carbon concentrations and

100 water flow in a nested catchment approach to quantify the effect of land management on aquatic carbon

101 concentrations and export. Specifically, we compare concentrations and speciation of aquatic carbon from

102 across three catchment types (non-drained, drained and restoration) to answer the following questions:

103 • How do land management practices across the Flow Country blanket bog affect aquatic carbon

104  concentrations, and how does this vary by carbon species?

105 • Is there evidence to suggest that aquatic carbon concentrations and fluxes from the restoration site are in

106  an intermediate state between drained (disturbed) peatland and non-drained (near-natural) peatland?

107 **2. Methods**

108 **2.1 Site description**

109 The study catchments are located c. 5 km northwest of Forsinard, northern Scotland, UK. Three study

110 catchments were identified within close proximity to represent three types of land management: non-drained,

111 drained (>40% of total catchment area affected by artificial drainage) and restoration (blocking of artificial

112 drains). Within each catchment, two stream monitoring sites were selected, splitting the experimental design

113 into six nested sub-catchments (Figure 1).

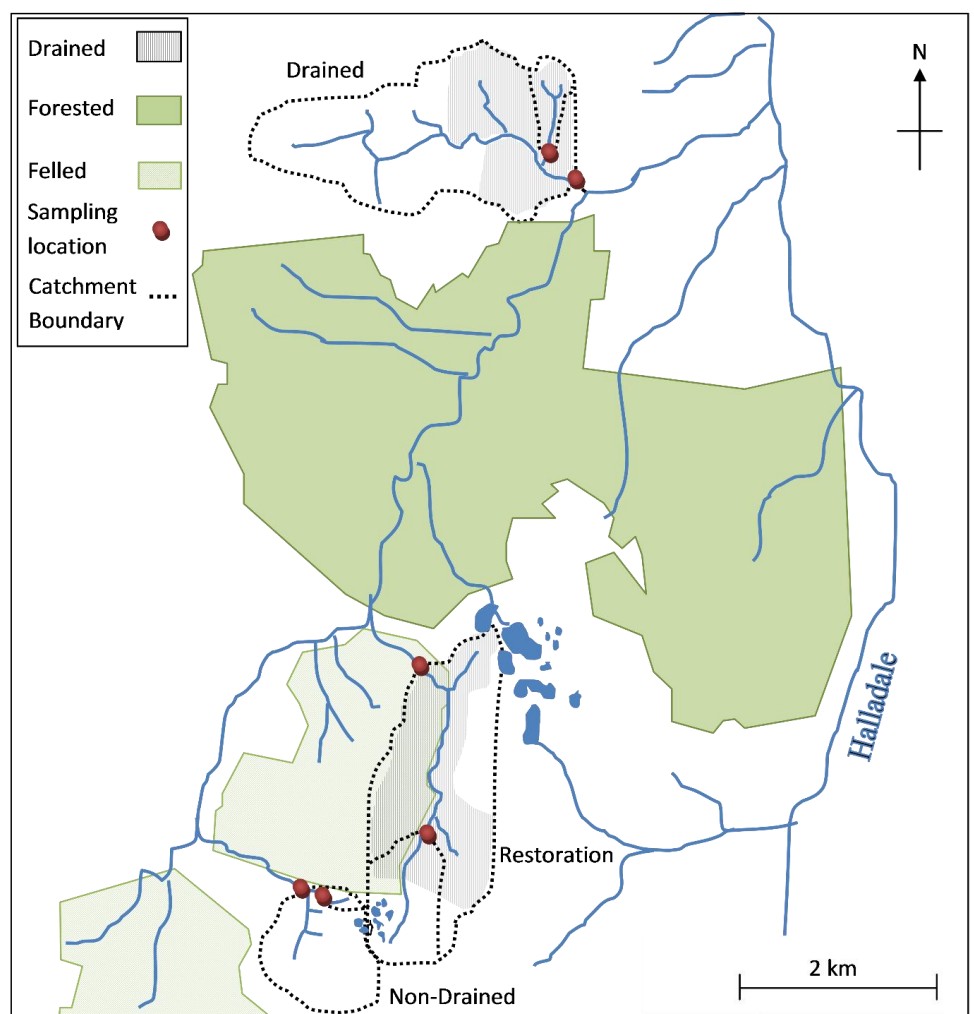

114

**Figure 1.** Schematic of experimental catchments including three land management types (Non-Drained, Drained and Restoration) and 2 nested sub-catchments (Upper and Lower). The diagram centre point has coordinates 58°24.45'N 3°56.80'W.

Both the non-drained and restoration catchments are located in the Cross Lochs area of the Royal Society for the Protection of Birds (RSPB)'s Forsinard Flows National Nature Reserve, while the restoration catchment forms part of the Bighouse Estate. The area has a mean annual temperature of 7.5 – 8.0 °C with a mean annual



precipitation range of 650 – 1000 mm. The geology consists of Moine granulites and schists over-laid with
fluvio-glacial material and blanket peat. Vegetation is dominated by mosses including *Sphagnum* spp. and
*Racomitrium lanuginosum* (Hedw.) Brid., sedges such as *Eriophorum* spp. and shrubs *Calluna vulgaris* (L.)
Hull and *Erica tetralix* L. Vegetation in the stream riparian zones is dominated by sedges and *Juncus*
*squarrous*.
The drains in Cross Lochs are believed to have been created in the 1970s and 1980s when farm capital grants
were made available. Areas of Cross Lochs were then planted in the early 1980s with non-native conifer
species (*Pinus contorta* and *Picea sitchensis*) (Lindsay et al., 1988). The RSPB began restoration of the area
in 2002 through the felling of trees and blocking of drains. At the time, given that the trees were still small,
trees were felled-to-waste, i.e. cut at the base and rolled into adjacent furrows. Drains of open ditch formation
were created on the Bighouse Estate during the 1950s in response to agricultural subsidies, and have been
regularly maintained and free flowing since their installation. In the lower catchment, drains are spaced
between 30 - 70 m apart; in the upper catchment, drains are spaced closer at approximately 30 - 40 m apart.
The study sites are small headwater streams of order 1 or 2 draining catchments ranging in size from 0.13 to
3.58 $km^2$ (Table 1). Whilst neither of the non-drained sub-catchments were affected by artificial drainage
alone, approximately 20% of the upper sub-catchment area has been influenced by forest-to-bog restoration.
The two drained sub-catchments contain no forestry or forest-to-bog restoration influence but have 65% and
25% of their total area affected by active artificial drainage (upper and lower sub catchments, respectively).
The restoration sub-catchments contain both forest-to-bog restoration and drain-blocking activity, with 40%
and 82% of the total area affected by blocked drains in the upper and lower restoration sub-catchments,
respectively.





**Table 1.** Sub catchment details.

|  | **Non-Drained** | | **Drained** | | **Restoration** | |
|---|---|---|---|---|---|---|
|  | Upper | Lower | Upper | Lower | Upper | Lower |
| **Acronym** | $N_U$ | $N_L$ | $D_U$ | $D_L$ | $R_U$ | $R_L$ |
| **Catchment size (km$^2$)** | 0.13 | 1.03 | 0.21 | 3.58 | 0.73 | 2.93 |
| **Area affected by open drains (%)** | 0 | 0 | 65 | 25 | 0 | 0 |
| **Area affected by blocked drains (%)** | 0 | 0 | 0 | 0 | 40 | 82 |
| **Tree removal (%)** | 20 | 0 | 0 | 0 | 32 | 19 |
| **Stream order** | 1˚ | 2˚ | 1˚ | 2˚ | 1˚ | 2˚ |
| **Elevation (m)** | 201 | 192 | 106 | 103 | 189 | 182 |


**2.2 Field sampling**
Stream water sampling was carried out approximately fortnightly over a two-year period from September 2008
to August 2010. On each sampling occasion and at each sampling point, a water sample was collected in a 500
mL acid-washed glass bottle for analysis of POC, DOC and DIC and a headspace and ambient air sample
collected in gas-tight syringes for analysis of $CO_2$ and $CH_4$. Stream water pH, temperature and electrical
conductivity (EC) were also measured using hand-held devices *in-situ* on each sampling occasion.
Stream height was continuously monitored throughout the full study period using pressure transducers (In-
Situ® Level TROLL®) positioned at the non-drained lower ($N_L$), drained lower ($D_L$) and restored upper ($R_U$)
stream sampling sites. These locations were chosen for their natural and stable conditions. Continuous
discharge was calculated using stage-discharge rating curves ($r^2$ between 0.84 and 0.97) created from dilution
gauging measurements correlating discharge at each individual sampling site to the catchment specific pressure
transducer.



**2.3 Laboratory analyses**
Stream water samples were filtered within 24 hours of collection through pre-ashed (6 hours at 500ºC), pre-
weighed Whatman GF/F (0.7 µm pore size) filter papers. POC was calculated using loss-on-ignition, following
the method of Ball (1964). The filtrate was stored in the dark at 4°C until analysis within four weeks of
sampling. The filtrate was analysed for DOC and DIC concentration using a PPM LABTOC Analyser with
detection range 0.1 to 4000 mg $L^{-1}$.
Dissolved $CO_2$ and $CH_4$ were calculated using the widely cited headspace technique (Billett et al., 2004;
Dinsmore et al., 2013; Kling et al., 1991). A 40 mL water sample was equilibrated with 20 mL of ambient air
at stream temperature by shaking vigorously under water for one minute; the equilibrated headspace was then
transferred to a gas tight syringe until analysis. On each sampling occasion a separate sample of ambient air
was also collected. Headspace samples were analysed on an HP5890 Series II gas chromatograph (Hewlett-
Packard), with flame ionisation detectors (with attached methaniser) for $CH_4$ and $CO_2$. Detection limits for
$CO_2$ and $CH_4$ were 10 ppmv and 70 ppbv, respectively. Concentrations of $CO_2$ and $CH_4$ dissolved in the stream
water were calculated from the headspace and ambient concentrations using Henry's law (e.g. Hope et al.,
2001). Although dissolved gaseous $CO_2$ and $CH_4$ form part of the DIC pool, due to the different measurement
methods employed here they are treated independently from DIC throughout this study, allowing comparison
with previous studies of peatland carbon budgets where this distinction has been made (e.g. Dinsmore et al.,
2010; Worrall et al., 2003).
**2.4 Data analysis**
One-way analysis of variance (ANOVA) was used to test differences in species specific carbon concentrations
between sampling sites, and significant differences were detected using a 95% confidence interval. To
determine the differences between individual groups, a post-hoc Tukey's test was applied to the ANOVA
results. Honestly significant differences were then reported using letters, where common letters indicate
statistically similar groups.
Carbon species concentration and discharge data were used to calculate the flow weighted mean concentration
(FWMC) following Equation 1 (Dinsmore et al., 2013), where $c_i$ is the instantaneous concentration, $q_i$ is the
instantaneous discharge and $t_i$ is the time step between concentration measurements.



$$\text{FWMC} = \frac{\Sigma(c_i \times t_i \times q_i)}{\Sigma(t_i \times q_i)} \quad (1)$$


Drivers of variability in the carbon FWMC were explored in multiple linear regressions using a step-wise
approach to construct a best-fit predictive model based on catchment land use data. Linear regression analyses
of carbon species data by site against air temperature and the natural log of discharge produced $r^2$ values and
p-values; these were then used to determine the strength and statistical significance of the relationships,
respectively. These analyses were conducted in R v 3.5.3 (R Core Team, 2018).
In order to reconcile the approximately fortnightly carbon concentration measurements with the continuous
discharge data to calculate annual carbon export, 'Method 5' of Walling and Webb (1985) was used, also
described in Dinsmore et al. (2013) and Hope et al. (1997). The method is shown in Equation 2, where $C_i$ is
the instantaneous concentration for each carbon species, $Q_i$ is the instantaneous discharge, $Q_r$ is the mean
discharge over the study period and $n$ is the number of instantaneous samples analysed.

$$\text{Load} = K \times Q_r \times \frac{\sum_n^{i-1}[C_i \times Q_i]}{\sum_n^{i-1} Q_i} . \quad (2)$$


Standard error of the load was derived using Equation 3, where $F$ is the annual discharge and $C_F$ is the flow-
weighted mean concentration (Hope et al., 1997).

$$\text{SE} = F \times \text{var}(C_F) . \quad (3)$$


The variance of $C_F$ was estimated using Equation 4, where $Q_n$ is the sum of all the individual $Q_i$ values (Hope
et al., 1997).

$$\text{var}(C_F) = \left[ \sum (C_i - C_F)^2 \times Q_i/Q_n \right] \times \sum Q_i^2/Q_n^2 . \quad (4)$$


Export values for each of the carbon species are reported in g m$^{-2}$ yr$^{-1}$ scaled to the catchment areas reported in
Table 1.





## 3. Results

### 3.1 Carbon concentrations

The concentration of DOC represented the greatest proportion of the total aquatic carbon component at all sites with mean concentrations ranging from a low of 12.8 mg C L$^{-1}$ in the upper non-drained catchment to a high of 20.5 mg C L$^{-1}$ in the upper drained catchment (Figure 2). Significant differences in DOC concentrations across the sampling period were observed between the upper non-drained catchment compared to the upper restoration catchment and both drained catchments (Table 2).

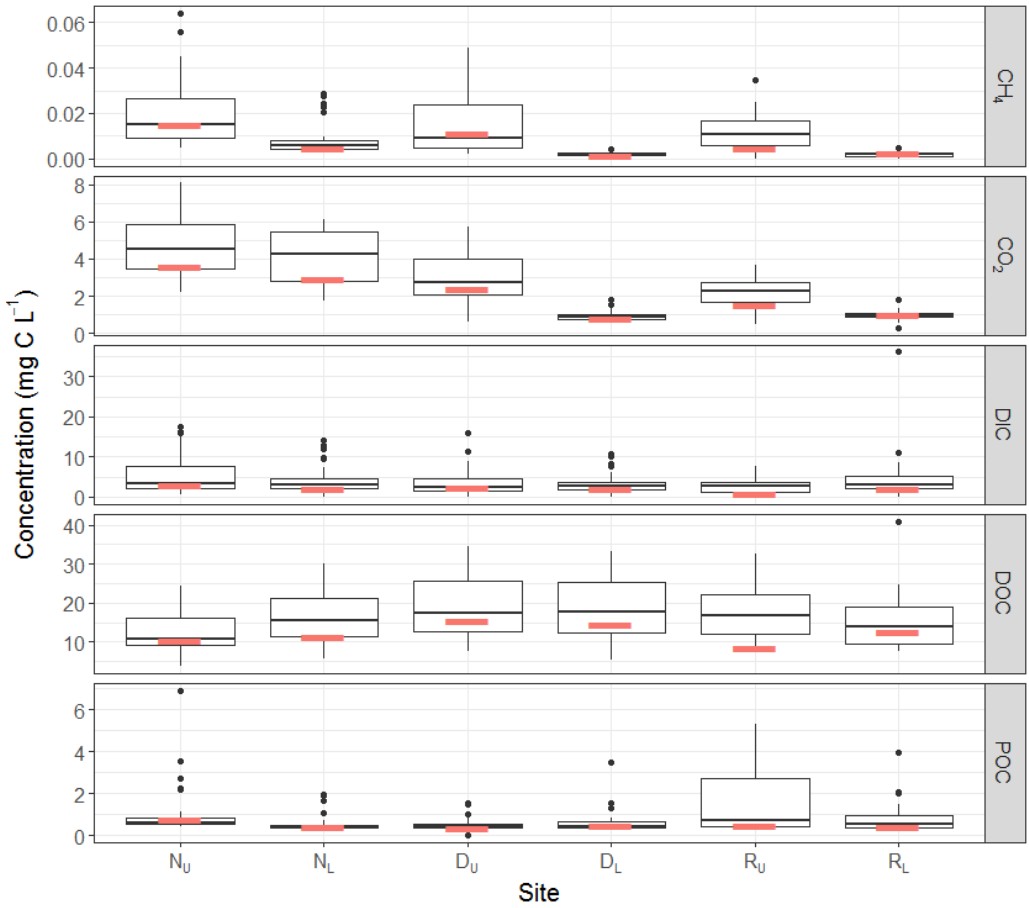

**Figure 2.** Boxplots showing range of carbon concentrations by species at each site over full measurement period, where the red line represents the flow weighted mean concentration.





The non-drained catchment had the greatest mean concentration of $CO_2$ at both the upper and lower sampling
sites, reaching a maximum of 8.1 mg C L$^{-1}$ (Table 2). Concentrations of $CO_2$ in the drained and restored
catchments were strongly dependent on sampling location, with concentrations at the upper sites greater than
those downstream, and this difference was significant for drained and restored catchments (Table 2). A similar
pattern was seen in the FWMCs suggesting this is more than a simple dilution effect (Figure 4). DIC
concentrations were of a similar magnitude to $CO_2$ at both the non-drained sub-catchments, but were
considerably higher than $CO_2$ in the drained and lower restored catchments.
**Table 2.** Mean (range) stream water hydrochemical data. * indicates gauged water level monitoring sites.
Letters in italics represent the results from Tukey's family test statistic with common letters indicating
statistically similar groups, as tested for each C species across sampling sites.

| | Non-Drained | | Drained | | Restoration | |
|---|---|---|---|---|---|---|
| | Upper | Lower* | Upper | Lower* | Upper* | Lower |
| Discharge (L s$^{-1}$) | 1.97 (0.37-16.03) | 15.81 (<0.01-154.34) | 7.39 (1.48-33.93) | 129.34 (5.3-686.44) | 32.51 (<0.01-300.69) | 64.14 (2.42-573.25) |
| $CO_2$ (mg C L$^{-1}$) | 4.64 (2.17-8.08) *a* | 4.23 (1.75-6.13) *a* | 2.97 (0.61-5.74) *b* | 0.98 (0.52-1.83) *d* | 2.24 (0.47-3.66) *c* | 0.97 (0.29-1.77) *d* |
| $CH_4$ (µg C L$^{-1}$) | 20.28 (4.49-63.87) *a* | 8.38 (2.49-28.76) *cd* | 17.32 (1.75-48.73) *ab* | 2.04 (0.7-4.15) *d* | 12.57 (0.04-34.94) *bc* | 1.74 (<0.01-4.66) *d* |
| DOC (mg C L$^{-1}$) | 12.82 (3.81-24.42) *a* | 17.73 (5.69-35.06) *ab* | 20.45 (7.53-42.19) *b* | 19.7 (5.49-33.13) *b* | 19.06 (8.19-36.34) *b* | 16.24 (7.53-40.96) *ab* |
| DIC (mg C L$^{-1}$) | 5.72 (0.7-17.61) *a* | 4.49 (0.04-14.09) *a* | 4.00 (<0.01-15.84) *a* | 3.82 (<0.01-10.82) *a* | 2.89 (<0.01-7.6) *a* | 4.64 (<0.01-36.08) *a* |
| POC (mg C L$^{-1}$) | 1.18 (0.39-6.93) *ab* | 0.59 (0.24-1.96) *a* | 0.56 (<0.01-1.51) *a* | 0.65 (0.24-3.47) *a* | 1.66 (0.34-5.34) *b* | 0.84 (0.21-3.96) *a* |
| Total C (mg C L$^{-1}$) | 24.38 | 27.05 | 28.00 | 25.15 | 25.86 | 22.69 |




Mean site CH$_4$ concentrations ranged from 1.7 µg C L$^{-1}$ at the lower restoration site to 20.3 µg C L$^{-1}$ in the
outflow of the upper non-drained catchment (Table 2). Within each site ranges were extremely high with the
maximum recorded concentration 63.9 µg C L$^{-1}$ at the upper non-drained catchment during Autumn 2009
(Figure 3). POC was also highly variable within catchments following a temporal pattern of low baseline
concentrations with sporadic peaks (Figure 3). Significantly higher POC concentrations were observed for the
upper restoration catchment (Table 2).

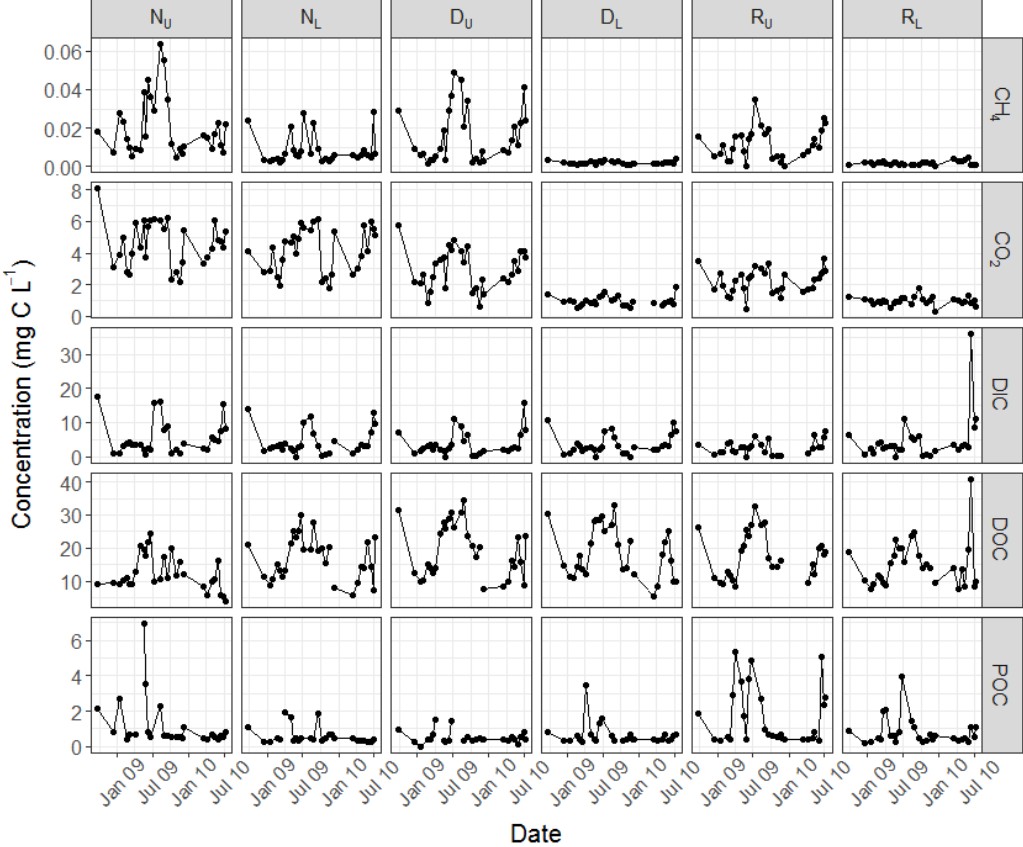


**Figure 3**. Time series of carbon concentrations by species across the six sampling sites.
Whilst the speciation of carbon was highly variable between catchments (Figure 5) with a number of between-
site significant differences at species level (Table 2), the site-specific mean total carbon concentrations were
all within the narrow range of 22.7 mg C L$^{-1}$ (R$_L$) to 28.0 mg C L$^{-1}$ (D$_U$).



Linear regression models were constructed with the aim of explaining the described site specific differences
in carbon concentrations based on catchment characteristics including total area, percent of catchment drained,
percent of catchment with blocked drains and percent of catchment that had undergone tree removal. When
single variables were included only total catchment area correlated significantly with $CO_2$ and $CH_4$ FWMCs;
no significant relationships existed for POC, DOC or DIC. Whilst not significant, the proportion of the
catchment that had been drained explained 58% of the site variation in $CO_2$ FWMC (p = 0.08, negative
relationship) and the proportion of the catchment that contained blocked drains explained 54% of the between
site variation in DOC FWMC (p = 0.09, positive relationship). These were the only other variables that had p-
values of less than 0.10.
Multiple linear regressions were then applied using a step-wise selection process that produced explanatory
models with p < 0.10 for $CH_4$, $CO_2$ and DOC (Table 3). High FWMCs of $CH_4$ were associated with sites that
contained few blocked drains and areas of tree removal. However as these variables themselves are correlated,
with blocked drains and tree removal occurring simultaneously, it is difficult to draw process-based
conclusions from these results. The $CO_2$ model suggests an increase in the drained area leads to lower stream
water concentrations; this is also seen in the DIC model that was non-significant. Catchments affected by tree
removal showed greater DIC concentrations. Given the inter-correlation between drain blocking and tree
removal at our test catchments, the positive relationship between $CO_2$ concentrations and blocked area may be
due to the same drivers as DIC and tree removal area.





**Table 3**. Best fit model describing between site variability in carbon FWMC based on stepwise multiple
linear regressions. Log10 transformation was applied to $CH_4$ FWMC before regressions were carried out.

| Species | Variables | Sign of relationship | $r^2$ | p-value |
|---|---|---|---|---|
| *$CH_4$* | Blocked Area | - | 0.87 | 0.02 |
| | Deforested Area | + | | |
| | | | | |
| *$CO_2$* | Total Area | - | 0.84 | 0.09 |
| | Blocked Area | - | | |
| | Drained Area | - | | |
| | | | | |
| *DOC* | Total Area | + | 0.69 | 0.08 |
| | Deforested Area | + | | |
| | | | | |
| *DIC* | *No model found* | --- | --- | --- |
| *POC* | *No model found* | --- | --- | --- |




Concentrations in all carbon species varied throughout the year (Figure 3). The majority of species, across all
sites, followed a seasonal pattern that positively correlated with air temperature (Table 4). Only DOC in the
upper non-drained and $CO_2$ in the lower restoration site did not display a positive relationship with average
daily air temperature. Temporal variability in carbon concentrations were also strongly linked to discharge,
primarily with a negative concentration-discharge relationship (Table 4). Only $CH_4$ concentrations in the lower
restored catchment showed a positive concentration-discharge relationship, and this was not significant at the
0.05 confidence interval.
**Table 4.** Results from linear regressions of concentration against log discharge and air temperature. Values
represent modelled $r^2$ values with †, * and ** representing p-values of <0.10, <0.05 and <0.01, respectively;
"ns" denotes non-significance at p > 0.10. +/- represents the sign of the relationship where one exists.

| Species | $N_U$ | $N_L$ | $D_U$ | $D_L$ | $R_U$ | $R_L$ |
|---|---|---|---|---|---|---|
| *Log(Discharge)* | | | | | | |
| $Log(CH_4)$ | - 0.2 * | - 0.28 ** | - 0.62 ** | - 0.58 ** | - 0.31 ** | + 0.11 † |
| $CO_2$ | - 0.44 ** | - 0.34 ** | - 0.71 ** | - 0.49 ** | - 0.54 ** | Ns |
| DIC | - 0.15 * | ns | - 0.37 ** | - 0.33 ** | - 0.34 ** | - 0.13 † |
| DOC | - 0.15 * | - 0.19 * | ns | ns | - 0.14 † | Ns |
| POC | ns | - 0.32 ** | - 0.13 † | - 0.11 † | - 0.55 ** | - 0.20 * |
| | | | | | | |
| *Air Temperature* | | | | | | |
| $Log(CH_4)$ | + 0.06 ** | + 0.14 ** | + 0.18 ** | + 0.03 ** | + 0.08 ** | + 0.02 ** |
| $CO_2$ | + 0.08 ** | + 0.18 ** | + 0.15 ** | + 0.09 ** | + 0.14 ** | |
| DIC | + 0.11 ** | + 0.14 ** | + 0.19 ** | + 0.13 ** | + 0.07 ** | + 0.03 ** |
| DOC | ns | + 0.14 ** | + 0.15 ** | + 0.05 ** | + 0.19 ** | + 0.05 ** |
| POC | + <0.01 * | + 0.03 ** | + 0.17 ** | + 0.10 ** | + 0.17 ** | + 0.20 ** |

**3.2 Hydrology**
Temporal hydrological regimes were similar among catchments with multiple 'flashy' storm peaks occurring
across all seasons. Peak flows were concurrent in time at all gauged streams (Figure 4). The drained site had
the highest mean (129 L s$^{-1}$) and peak discharge (686 L s$^{-1}$), compared to non-drained or restoration sites that
had discharge means of 15 L s$^{-1}$ and 32 L s$^{-1}$, respectively. Since the gauged catchments cover a range of
upstream catchment areas (Table 1), it is, therefore, potentially more useful to compare runoff values (Table
2). Of the gauged sites, annual runoff was greatest from the restoration site (1404 mm), followed by the drained
(1139 mm) and the non-drained sites (475 mm), respectively. The annual runoff for both the upper and lower
sites in the non-drained and drained catchments were very similar, however runoff at the upper site was more



than double that at the lower site in the restoration catchment with values of 1404 mm and 679 mm,
respectively. The two restoration sub-catchments also differed significantly in the percent of the catchment
that is affected by blocked drains (upper 40%, lower 82%).

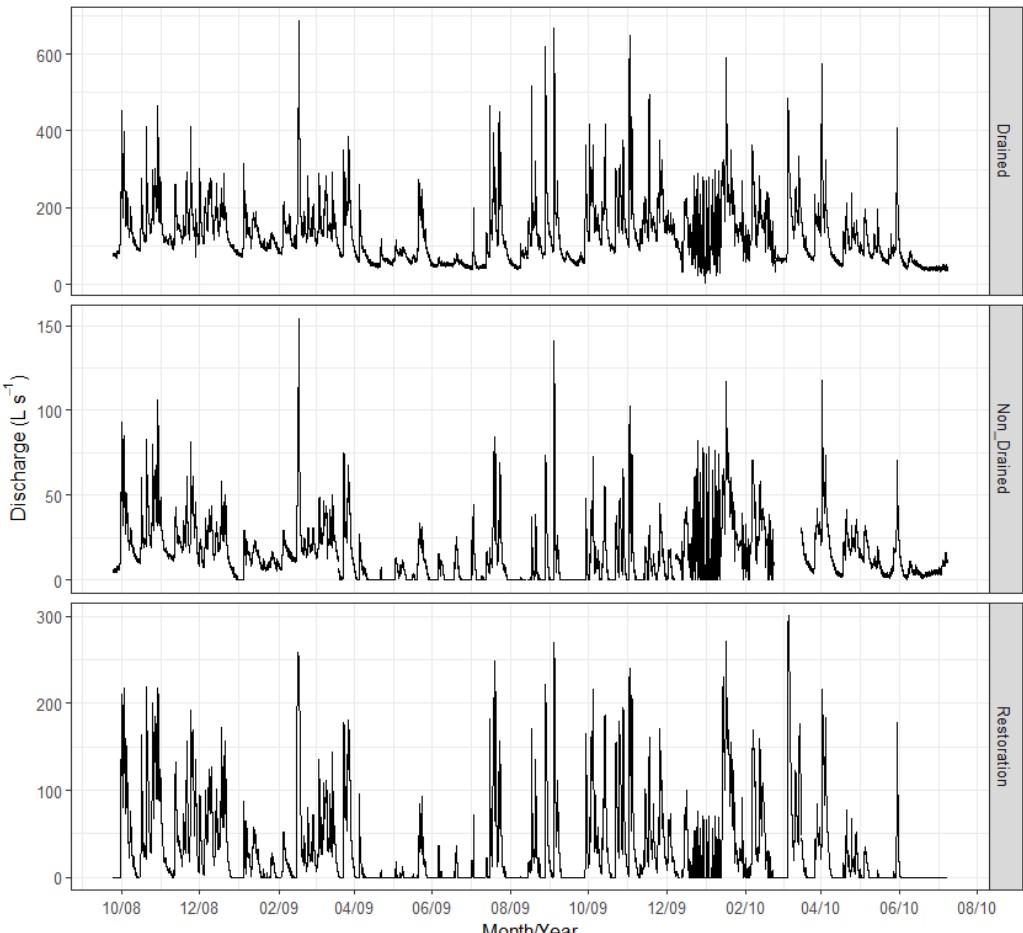


**Figure 4.** Discharge time series from pressure transducers located at sites $N_L$, $D_L$, $R_U$, representing the Non-
Drained, Drained and Restoration catchments, respectively.
The gauged site in the non-drained catchment displayed the steepest flow duration curve indicating high flows
lasting the shortest periods (Figure 5); this is most likely a result of the small catchment size rather than an
indication of the water holding capacity. Despite a much larger upstream catchment area, the drained site also
displayed a steep curve, with the shallowest curve at the upper flow limit displayed by the restoration



catchment. The base flow contributions follow the expected distribution based on catchment size (drained >
non-drained > restoration).

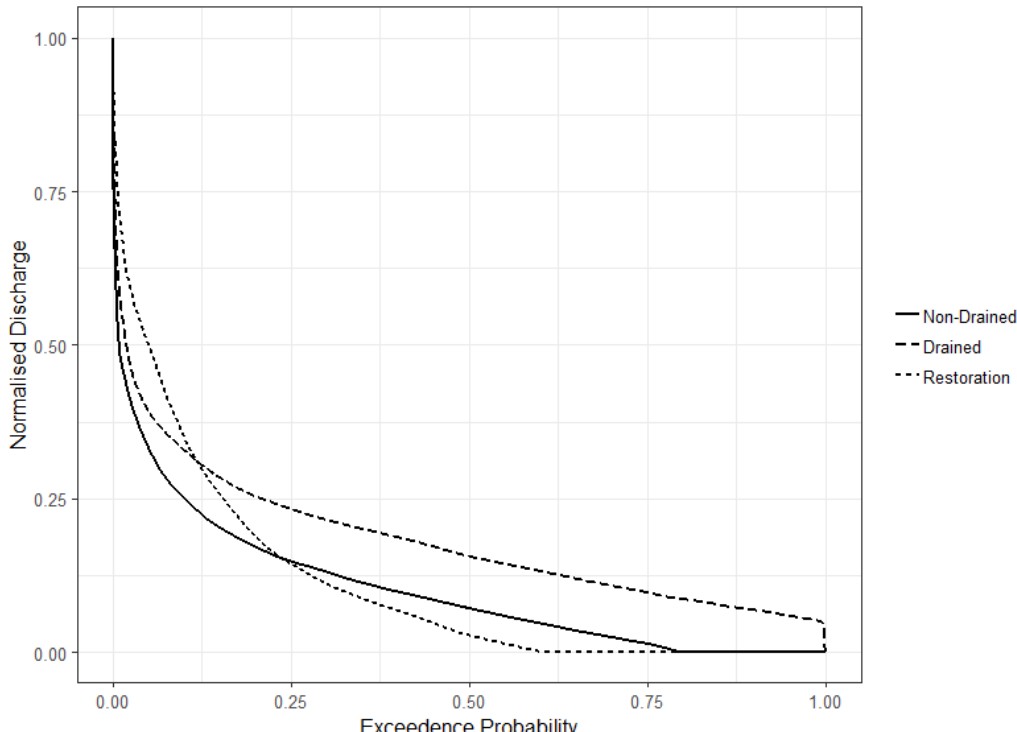


**Figure 5.** Flow duration curve showing exceedance probability of normalised discharge across the three
gauged sites.
**3.3 Carbon Export**
Only downstream fluvial carbon export is calculated in this study, therefore, the results below do not take
account of aquatic exports via the vertical evasion of dissolved gases from the water surface. The greatest total
fluvial carbon exports were measured in the two drained sites (26.7 and 24.6 g C m$^{-2}$ yr$^{-1}$ for the upstream and
downstream catchments, respectively); the smallest measured total exports were for the two non-drained sites
(10.0 and 10.8 g C m$^{-2}$ yr$^{-1}$ for the upstream and downstream catchments, respectively; Table 5).



**Table 5.** Downstream carbon export for each catchment ± SE over full study period in g C m$^{-2}$ yr$^{-1}$.

| | $N_U$ | $N_L$ | $D_U$ | $D_L$ | $R_U$ | $R_L$ |
|---|---|---|---|---|---|---|
| $CH_4$ | 0.007 ± < 0.001 | 0.002 ± < 0.001 | 0.014 ± < 0.001 | 0.002 ± < 0.001 | 0.006 ± < 0.001 | 0.002 ± < 0.001 |
| $CO_2$ | 1.81 ± 0.04 | 1.49 ± <0.01 | 2.77 ± 0.02 | 0.91 ± <0.01 | 2.00 ± <0.01 | 0.69 ± <0.01 |
| DIC | 2.15 ± 0.31 | 1.60 ± 0.03 | 3.25 ± 0.10 | 3.04 ± <0.01 | 2.10 ± 0.01 | 1.46 ± 0.01 |
| DOC | 5.62 ± 0.44 | 7.56 ± 0.10 | 20.16 ± 0.63 | 19.98 ± 0.04 | 18.40 ± 0.08 | 8.94 ± 0.02 |
| POC | 0.44 ± 0.02 | 0.18 ± <0.01 | 0.53 ± <0.01 | 0.62 ± <0.01 | 0.75 ± <0.01 | 0.32 ± <0.01 |


Whilst variability between the nested sub-catchments at the non-drained and drained sites was very low, the
two sub-catchments in the restored area varied significantly from a total carbon export of 23.3 g C m$^{-2}$ yr$^{-1}$ at
the upper site to 11.4 g C m$^{-2}$ yr$^{-1}$ at the lower site (Figure 6). The species which contributed most to the total
fluvial carbon export was DOC across all catchments, with the second most important export component DIC
followed by $CO_2$. POC fluxes were an order of magnitude lower than DIC fluxes, and export of $CH_4$ was minor
across all catchments.

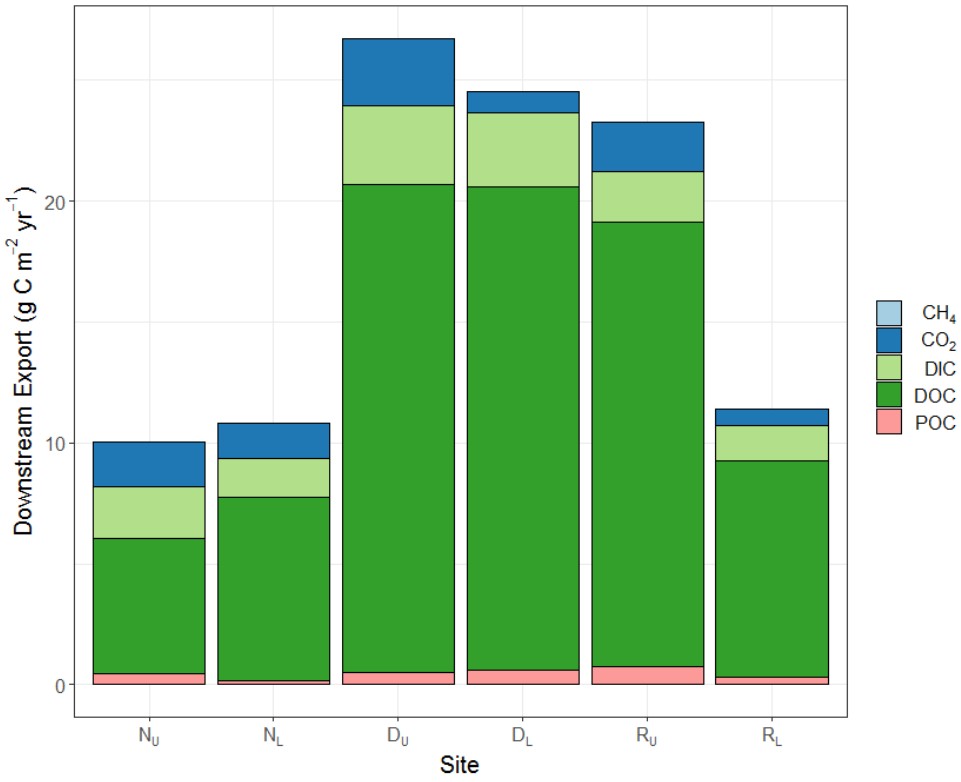


**Figure 6.** Total downstream carbon export from each site separated by carbon species.





## 4. Discussion

### 4.1 Carbon concentrations under different peatland land management

This study provides the first assessment of concentrations of all waterborne carbon species (including $CO_2$ and $CH_4$) in small headwater catchments located in the Flow Country and will provide a reference point for future comparisons of these systems, particularly as they respond over the long-term to management. Under all peatland land management types DOC was the largest component of total aquatic carbon. Concentrations were within the range measured in previous studies of blanket bogs (Evans et al., 2018; de Wit et al., 2016) and followed the typical seasonal cycle observed in peatlands, where concentrations tend to peak during late summer/early autumn (Figure 3). Highest mean concentrations were observed in the drained catchment. Previous studies in the Flow Country have indicated that stream DOC concentrations increase in the short-term following peatland restoration interventions, in part due to the disturbance of the land (Shah and Nisbet, 2019; Gaffney et al., 2020), yet this effect was not detected here. Time since intervention may have subdued the effect of restoration on DOC concentration, as measurements were started approximately six years after restoration work began in the area. It should be noted that in a 17-year-old forest-to-bog restoration site also located within the Flow Country, mean DOC concentrations remained ~ two fold higher than non-drained bog sites in both surface- and pore-water (Gaffney et al., 2018), suggesting that these effects can be detected over the longer timescales. Potential drivers of variability between the findings of this study and Gaffney et al. (2018) include percentage of catchment area affected by restoration works and the scale of investigation (plot scale versus catchment scale).

POC concentrations were relatively low across all sites, and there was little evidence of drainage increasing concentrations, as has been observed in highly degraded peatlands in the UK (Pawson et al., 2012; Yeloff et al., 2005). This suggests that the ditches in the drained catchment were not actively eroding at the time of this study or that our fortnightly sampling interval did not capture peak flows when increased POC export might be expected, although no positive POC-discharge relationships were observed at the sampling sites in this study (Table 4). Peatland disturbances other than drainage can also contribute to short-term increases in POC concentrations (Heal et al., 2020; Nieminen et al., 2017) and a significant difference was detected for concentrations in the upper restoration catchment, which, in percentage coverage terms, was most affected by forest-to-bog restoration (Table 1). The technique of fell-to-waste, whereby tree material is left on-site post-



restoration, was utilised in the Cross Lochs area, and this may have contributed to the observed POC effect.
The degree to which sediment traps put in place as part of the drain blocking process during forest-to-bog
restoration are effective at capturing POC (Andersen et al., 2018) requires further testing.
Concentrations of dissolved $CO_2$ were highest in the non-drained catchments, although the degree to which
this can be attributed to peatland land management is uncertain. Whilst increased $CO_2$ partial pressures have
similarly been found in undrained catchments compared to drained catchments in a Finnish peatland (Rantakari
et al., 2010), a more likely explanation in this study is that total catchment area was the dominant driver of
dissolved $CO_2$ concentrations, as detected in multiple linear regression modelling (Table 3). Concentrations
were consistently higher in the upper catchments of all land management types, with significant differences
observed in the drained and restoration sub catchments. Low order streams in small catchments inherently
have a higher degree of connectivity with the surrounding peatland soil, resulting in $CO_2$ supersaturation
(Wallin et al., 2010). Rapid evasion of supersaturated $CO_2$ from headwater peatland streams has been widely
observed (Billett et al., 2015; Hope et al., 2004; Kokic et al., 2015), and is suggestive that the differences
detected in this study could, at least in part, be attributed to evasion during transit between first and second
order streams. That the lowest difference in $CO_2$ concentration was detected in the non-drained catchment
where there was the smallest distance between upper and lower sampling points (Figure 1) further supports
this proposition. Evasion of $CO_2$ in headwaters may be a significant component of peatland carbon budgets
and should be quantified as a specific loss term, particularly when isotopic analyses have determined the
evaded $CO_2$ to be 'young', and therefore intrinsically related to the peatland's contemporary net ecosystem
carbon balance (Billett et al., 2015).
Dissolved $CH_4$ concentrations followed the same trend as $CO_2$: highest concentrations were consistently
detected in the upper catchments. Several studies have examined $CH_4$ emissions in peatlands where water
tables have been artificially raised through ditch blocking and suggest that infilled drains may be acting as "hot
spots", particularly when the presence of species with aerenchyma such as Eriophorum angustifolium allows
$CH_4$ to bypass oxidative pathways (Cooper et al., 2014; Günther et al., 2020; Waddington and Day, 2007), but
comparatively fewer studies have looked at dissolved $CH_4$ in streams receiving water from peatlands.
However, in a study of dissolved $CO_2$ and $CH_4$ concentrations in blocked and open ditches in a blanket bog in
N Wales with a higher level of experimental replication than in this study, there was no evidence of systematic

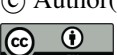



differences between the two ditch types (Evans et al., 2018). Similarly, there was no evidence of this effect in
the catchments monitored in this study and concentrations were similar to those detected by Evans et al. (2018).
While the lack of detection of a land management effect is perhaps unsurprising as a consequence of the low
experimental replication and time since intervention, it may also relate to multiple controls (organic matter,
terminal electron acceptors, hydrology, geomorphology, etc.) that operate in relation to methane production
and processing in streams, which remain poorly understood (Stanley et al., 2016).
**4.2 Effects of peatland land management on flow regimes**
Flow regimes varied considerably between the six monitoring sites included in this study. Increased annual
runoff was detected in the drained catchments (mean: 1125 mm) relative to the non-drained catchments (mean:
471 mm), suggesting that peatland drainage has had a profound impact on catchment hydrological functioning.
Drainage of blanket peatland has previously been shown to modify flow pathways, via a shift from overland
flow to throughflow (Holden et al., 2006), and to increase peak flows (Ballard et al., 2012). Flow duration
curves indicated that peak flows lasted longer in the drained catchment relative to the non-drained catchment,
although it was in the restoration catchment where peak flows were sustained for the longest periods. This was
a surprising result, although it should be noted that the restoration catchment was the only land management
type where flow monitoring occurred at the upper rather than lower sampling point, and it was at this site that
highest catchment runoff was observed. Lack of pre-intervention data means that we are unable to assess
inherent differences in hydrology between the study sites, although the occurrence of periods of dry-out at both
the non-drained and restoration stream monitoring sites (Figure 4) suggests that there may be significant
movement of water out of the catchment via other flow paths (e.g. sub-surface or overland) which are not
quantified here.
Annual runoff for the two restoration sites was markedly different (Table 2), with the lower site's runoff similar
to the non-drained catchments, and the upper site's runoff exceeding that of the drained catchments.  There
was a large difference in the percentage of catchment area affected by restoration activities, with the lower
catchment affected by considerably more ditch blocking. It follows that water flux from the lower catchment
would be reduced, as has been discerned in other ditch-focussed studies of peatland restoration (Evans et al.,
2018). This has previously been attributed to an increase in evaporation relative to precipitation in restored
catchments, which occurs because water is retained in the catchment for longer, partly due to the physical

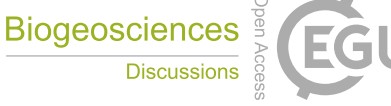

barrier that peatland ditch blocks create whereby water pools behind the peat or piling dams (Peacock et al.,
2013) and is more susceptible to evaporative loss. However, whilst this process may have had a small role in
contributing toward the observed runoff differences, its overall impact it likely to be limited in the northern,
temperate climate of the Flow Country, where high cloud cover, low temperatures and high contributions from
occult precipitations reduces potential for evaporation (Lapen et al., 2000).
Another potential explanation for the observed differences in runoff is that in areas affected by peatland
restoration works, a greater proportion of total runoff occurs as overland or near-surface flow (Holden et al.,
2017b). This flow can effectively bypass typical drainage networks and is therefore not necessarily represented
in the stream discharge data presented in this study. Previous studies have found diversion to overland flow to
explain the difference in runoff measured between restored and control peatland catchments (Holden et al.,
2017a; Turner et al., 2013). Although data were not collected here that can verify the contribution of different
flow paths to total catchment runoff, it is feasible that flow path shifts have been initiated in the lower
restoration catchment following ditch blocking. As clear differences in runoff are evident between the drained
and non-drained catchments, this could be interpreted as a signal of the successful hydrological restoration of
the lower catchment and its movement towards more natural functioning.
**4.3 Impacts of restoration on carbon fluxes**
Aquatic carbon fluxes from all catchments were within the same order of magnitude, although were
consistently lower than those detected in a previous study of all waterborne carbon species in a stream draining
from a peatland in southern Scotland, where DOC alone contributed to a flux of 25.4 g C m$^2$ yr$^{-1}$ (Dinsmore et
al., 2010). The fluxes were similar to those detected from headwater streams in the Flow Country (Gaffney et
al., 2020). Although the Gaffney et al., (2020) study did not measure $CO_2$ and $CH_4$, this did not lead to large
differences in carbon export between the studies, as DOC was the dominant flux term in both overall budgets.
However, $CO_2$ was the third largest contributor to total carbon export following DOC and DIC suggesting that
the dissolved gaseous component is important to include in total export estimates, particularly as it has potential
for rapid evasion and, therefore, influence on peatland greenhouse gas budgets.
The same catchment was employed as the non-drained lower catchment in this study (measurements from
2008 to 2010) and as the 'bog control' in the Gaffney et al. (2020) study (measurements from 2013-2015), and

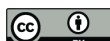



carbon fluxes here were notably lower (10.8 vs.18.4 g C m$^2$ yr$^{-1}$; mean of 2014 and 2015 C export). As there
is only a small difference in carbon concentrations between the studies, the difference is likely to be due to
inter-annual hydrological and climatic variation. This finding highlights the limitation of taking measurements
over only a few years, as it is well established that carbon export can vary considerably as a function of inter-
annual hydrological variation. The influence of varying hydrology, including precipitation and evaporation
balances, catchment water storage and flow path routing, may mask the potentially more subtle differences in
biogeochemistry, and associated carbon fluxes, that arise due to land management practices.
Aquatic carbon export varied between the land management types, and the drained and non-drained sites were
markedly different in their overall carbon flux, with average fluxes nearly 150% greater from the drained
catchments. This finding indicates the dramatic effect that drainage, particularly when maintained, can have
on peatland aquatic carbon fluxes or, at the very least, the dominant flow paths within a catchment, for example
open channel flow (as measured here) versus overland and sub-surface flow (not quantified here). There was
large intra-site variability in carbon fluxes within the restoration sub-catchments, which means it is difficult to
determine the impact of the restoration activities on aquatic carbon losses. The degree to which the nested
experimental design employed here can determine a confident land management effect on stream carbon
concentrations and fluxes is questionable.  The nested design limited true replication between the land
management types, and greater replication of all land types would be required to conclude that land
management alone was the driver of the observed differences. Furthermore assessment of restoration success
without prior monitoring of stream carbon is not optimal and a before-after-control-intervention approach is a
better experimental approach. Turner *et al*, (2008), examined stream DOC concentrations pre- and post-
restoration and demonstrated that without pre-restoration information, a different conclusion regarding the
success of restoration would have been reached. Thus, where practical, monitoring of pre-restoration
conditions should be attempted to give a more accurate assessment of restoration success, and this requires
active communication between researchers and land managers in order to ensure that monitoring is established
ideally at least one year before restoration interventions occur.
**5  Conclusions**
Our study measured all waterborne carbon species in streams draining from blanket bog in the Flow Country
in order to assess the effects of varying peatland land management. Increased dissolved organic carbon





concentrations were detected in areas of drained peatland relative to non-drained peatland, and there was
considerable variation in speciation of carbon across the monitoring sites. Aquatic carbon fluxes were
intrinsically linked to catchment hydrology, and large differences in runoff, particularly between the
restoration sites, generated uncertainty regarding the impact of peatland restoration on fluvial carbon losses.
We recommend that future studies combine detailed measurements of carbon speciation, as presented here,
with rigorous hydrological monitoring to quantify carbon losses via different catchment flow paths, before and
after peatland management interventions. With this approach the impact of peatland restoration on both aquatic
carbon concentrations and fluxes can be fully quantified.

## 6 Data Availability

Carbon concentration data for all sites are available via the Environmental Information Data Centre (Pickard
et al., 2021).

## 7 Author Contributions

MB collected field samples and undertook laboratory analyses. Data analysis was performed by KJD, AEP
and MB. MFB provided guidance on the scope and design of the project and RA contributed land
management data. AEP prepared the manuscript, with contributions from KJD, MFB and RA.

## 8 Acknowledgments

Field and laboratory work for this study was undertaken during a PhD studentship awarded to Marcella
Branagan, and was funded via the 'UHI Addressing Research Capacity' Project, with financial contributions
from the European Regional Development Fund, Highlands and Islands Enterprise and the Scottish Funding
Council. Manuscript preparation time was funded by the Natural Environment Research Council, UK, as part
of the Land Ocean Carbon Transfer (LOCATE; **http://locate.ac.uk**) project, grant number NE/N018087/1.
RA was funded by a Leverhulme Leadership Award (RL-2019-02) and access to land management data was
made possible through the NERC "How does land management influence FIre REsilience in BLANKET bogs
(FIRE BLANKET)" project, grant number NE/T006528/1. Thanks to both the RSPB and the Bighouse Estate
for facilitating access to the sampling sites across the project duration, and to the RSPB for confirming the
chronology of restoration activities within the Cross Lochs area.



**9    Competing interests**
The authors declare that they have no conflict of interest.

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
