# Peer review of "Effects of Peatland Management on Aquatic Carbon Concentrations and Fluxes"

_Biogeosciences, 2021_

## Author Comment (AC2)

**CC2 comment**

Some more comments:

0) Table 1 could do with some information on slope within each sub-catchment (as this will influence runoff and erosion and residence time). Maybe some information on this can be added?

1) The Figure 1 could be improved by chosing a different colour for the catchment outline of Drained, Restoration, Non-drained. Currently it is easy to overlook this (font could also be bigger and bold for the conditions (e.g. Drained).

2) To my knowledge the Ball (1964) LOI conversion has never been tested for POC from peatlands. Our comparison did not provide good results and C/N analysis provides a better alternative. Could this have impacted the results?

3) Table 4 correct NS to ns in last column.

4) Fig. 4 This is a nice graph but more informative would be to have the discharge expressed per unit of sub-catchment area (to allow a weighted comparison).

It is good to see the BACI approach discussed - we need more of it - and also the various C export aspects (although I have been wondering about how CO2 evasion from streams might actually be from aquatic organisms unrelated to the actual peat body and its C balance - they fix and respire C as well !).

5) Finally, I wonder about a mention in the discussion around the fate of the exported C. We really do not know yet how much of the DOC & POC will be 'lost' via stream and river transport. The results vary a lot and the measurements were often artificial (cuvettes), possibly not accurately mimicking temperature and light conditions (in stream/river conditions). Further resarch is needed ...

**Author reply**

Thank you for your helpful comments. Please find our responses as follows:

0)  We agree that this information is useful for interpretation and have added information to Table 1 on mean slope for each sub-catchment. The mean slopes are subsequently referred to in discussion in terms of erosion potential.
1)  We have amended the figure as specified.
2)  Loss on Ignition following the Ball (1964) conversion method has been used to detect POC concentrations from peatlands in previous studies (Dinsmore et al., 2013), yet it is acknowledged in Dinsmore et al. (2010) that "…given the generally low concentrations of POC in water samples, analytical error is acknowledged to be significant, runs containing deionised water in place of sample (blanks) produced an error of ~15% of the mean POC concentration". We have added a statement about the uncertainty introduced by this analytical method when it is introduced in the text.
3)  Corrected as specified.
4)  We have amended figure 4 to express the discharge per unit area.

We agree that examination of the origin of dissolved $CO_2$ (i.e. from soil supersaturation, geological sources, or from in-stream microbial activity) is an interesting research question. Isotopic analyses combined with 14-C dating may provide further elucidation, and in a past study conducted in UK and Finnish peatlands it was strongly suggested that a significant proportion of $CO_2$ lost from headwaters by evasion was *not* derived from within-stream breakdown of DOC, as evidenced by the difference in

the $\delta^{13}C$ and $^{14}C$ signatures of DOC and evasion $CO_2$ (Billett et al., 2015). However, variation between different peatlands is likely to be high, and these isotopic data would certainly be of interest for our Flow Country monitoring sites to help understand the origin of dissolved $CO_2$.

5) The fate of exported C is a very interesting research topic. All sub-catchments within this study drain into the Halladale, which is a short-residence time river with very high aquatic DOC fluxes at the tidal limit in the context of GB (Williamson et al., 2021). Given the short transit time of water within the freshwater continuum and lack of any standing water bodies within the catchment below these sampling stations where biogeochemical transformations are likely to occur (Anderson et al., 2019), it is probable that much of the carbon is transported conservatively until it reaches the estuarine environment. We have added information on the fate of C to the discussion to put our findings into a wider carbon balance context.

**References**

Anderson, T.R., Rowe, E.C., Polimene, L., Tipping, E., Evans, C.D., Barry, C.D.G., Hansell, D.A., Kaiser, K., Kitidis, V., Lapworth, D.J., Mayor, D.J., Monteith, D.T., Pickard, A.E., Sanders, R.J., Spears, B.M., Torres, R., Tye, A.M., Wade, A.J., Waska, H., 2019. Unified concepts for understanding and modelling turnover of dissolved organic matter from freshwaters to the ocean: the UniDOM model. Biogeochemistry. https://doi.org/10.1007/s10533-019-00621-1

BALL, D.F., 1964. LOSS-ON-IGNITION AS AN ESTIMATE OF ORGANIC MATTER AND ORGANIC CARBON IN NON-CALCAREOUS SOILS. J. Soil Sci. https://doi.org/10.1111/j.1365-2389.1964.tb00247.x

Billett, M.F., Garnett, M.H., Dinsmore, K.J., 2015. Should Aquatic CO2Evasion be Included in Contemporary Carbon Budgets for Peatland Ecosystems? Ecosystems. https://doi.org/10.1007/s10021-014-9838-5

Dinsmore, K.J., Billett, M.F., Dyson, K.E., 2013. Temperature and precipitation drive temporal variability in aquatic carbon and GHG concentrations and fluxes in a peatland catchment. Glob. Chang. Biol. https://doi.org/10.1111/gcb.12209

Dinsmore, K.J., Billett, M.F., Skiba, U.M., Rees, R.M., Drewer, J., Helfter, C., 2010. Role of the aquatic pathway in the carbon and greenhouse gas budgets of a peatland catchment. Glob. Chang. Biol. https://doi.org/10.1111/j.1365-2486.2009.02119.x

Williamson, J.L., Tye, A., Lapworth, D.J., Monteith, D., Sanders, R., Mayor, D.J., Barry, C., Bowes, Mike, Bowes, Michael, Burden, A., Callaghan, N., Farr, G., Felgate, S., Fitch, A., Gibb, S., Gilbert, P., Hargreaves, G., Keenan, P., Kitidis, V., Juergens, M., Martin, A., Mounteney, I., Nightingale, P.D., Pereira, M.G., Olszewska, J., Pickard, A., Rees, A.P., Spears, B., Stinchcombe, M., White, D., Williams, P., Worrall, F., Evans, C., 2021. Landscape controls on riverine export of dissolved organic carbon from Great Britain. Biogeochemistry 2. https://doi.org/10.1007/s10533-021-00762-2

---

## Author Comment (AC3)

**CC1 Comment**

Really nice study! Congratulate the authors on such a neat design and presentation of the findings.

I would like to suggest the authors include a clarification in the title as to the type of management. Only too often are papers quoted in relation to management (generic) whilst the actual study focused on one issue (drainage). So here it is addressing past afforestation (and drainage!) on peatlands.

**Author reply**

Many thanks for your positive comments.

Based on your feedback we can adjust the paper title to the following, if acceptable to the journal to change at point of review: "Effects of Peatland Drainage and Restoration on Aquatic Carbon Concentrations and Fluxes"

---

## Author Comment (AC4)

*Reviewer comment in italics*

Author response in plain text

**RC1 Comment**

*The manuscript represents a case study related to peatland management's influence on aquatic carbon concentrations and fluxes in the UK. Peatland restoration actions are of interest in scientific communities globally and especially in the northern hemisphere where peatlands have been traditionally been used for active land use purposes. In overall study is well written. All information about the influence of peatland management actions on carbon dynamics is important to document and share with the scientific community and land-use managers. Few further suggestions for authors to improve their manuscript and analysis*

Thank you for your positive comments about the paper and for your helpful suggestions to improve it.

*The authors state already in the abstract that long-term monitoring is needed. I fully agree with them and was wondering that since measurement for this study has been done already ~10 years ago (2008-2010) may authors have some new data to be added to the time series? This would strengthen results a lot and give also long-term perspective.*

Unfortunately, as with many studies, these data were collected over a relatively short time frame dictated by the associated PhD project's timeline. However there has been further monitoring of the Halladale catchment into which these sub-catchments drain (e.g. Williamson et al., 2021) and indeed of some of the same sub-catchments (Gaffney et al., 2020, 2018). We refer to the Gaffney et al. (2018, 2020) papers in the discussion to provide further context about our findings, and have since added a comparison with the Williamson et al., (2021) findings. These comparisons have highlighted that our measured fluxes from the non-drained sites are low and we further explore this as a function of dry-out in our paper revisions, as described in the next comment.

*Authors should use "specific discharge, l/s/km2" instead of discharge eg in fig 4. This would enable better comparison between the catchment as their catchment size varies. Also, non-drained and restoration sites dry out (no discharge) during several periods. Authors need more to discuss that how this is influencing their concentration and fluxes.*

We have amended figure 4 as recommended by the reviewer, and it now shows the specific discharge for each sub-catchment. We agree that the periods of dry out experienced in the non-drained and restoration sites, whilst referred to in the discussion, may be having a negative influence on the overall fluxes. We have stress-tested this effect by removing occurences of dry out (i.e. zero flow) from the carbon flux calculations and have added this comparison to the methodology to outline the overall effect on fluxes in percentage terms. This is discussed further in the overall interpretation and we highlight that dry-out is partly responsible for the comparatively low fluxes measured, particularly from the non-drained catchments.

*One of the author's main conclusions is that future studies should use a before-after-control experiment. I fully agree with the statement but would like to note that this procedure has been already implemented to monitoring programs over 10 years ago for example in Fenno-Scandinavian countries. Also, there are several studies done using the before-after-control-impact approach related to peatland restoration and authors should update their references.*

Upon reflection, we agree that our literature is too UK-focussed and have expanded our references in the discussion to include more international studies, with a focus on Fenno-Scandanavian studies employing a B-A-C-I experiment design, as per the reviewer's comment (e.g. Haapalehto et al., 2014; Menberu et al., 2017; Strack et al., 2015).

**References**

Gaffney, P.P.J., Hancock, M.H., Taggart, M.A., Andersen, R., 2020. Restoration of afforested peatland: Immediate effects on aquatic carbon loss. Sci. Total Environ. https://doi.org/10.1016/j.scitotenv.2020.140594

Gaffney, P.P.J., Hancock, M.H., Taggart, M.A., Andersen, R., 2018. Measuring restoration progress using pore- and surface-water chemistry across a chronosequence of formerly afforested blanket bogs. J. Environ. Manage. https://doi.org/10.1016/j.jenvman.2018.04.106

Haapalehto, T., Kotiaho, J.S., Matilainen, R., Tahvanainen, T., 2014. The effects of long-term drainage and subsequent restoration on water table level and pore water chemistry in boreal peatlands. J. Hydrol. https://doi.org/10.1016/j.jhydrol.2014.09.013

Menberu, M.W., Marttila, H., Tahvanainen, T., Kotiaho, J.S., Hokkanen, R., Kløve, B., Ronkanen, A.K., 2017. Changes in Pore Water Quality After Peatland Restoration: Assessment of a Large-Scale, Replicated Before-After-Control-Impact Study in Finland. Water Resour. Res. https://doi.org/10.1002/2017WR020630

Strack, M., Zuback, Y., McCarter, C., Price, J., 2015. Changes in dissolved organic carbon quality in soils and discharge 10 years after peatland restoration. J. Hydrol. https://doi.org/10.1016/j.jhydrol.2015.04.061

Williamson, J.L., Tye, A., Lapworth, D.J., Monteith, D., Sanders, R., Mayor, D.J., Barry, C., Bowes, Mike, Bowes, Michael, Burden, A., Callaghan, N., Farr, G., Felgate, S., Fitch, A., Gibb, S., Gilbert, P., Hargreaves, G., Keenan, P., Kitidis, V., Juergens, M., Martin, A., Mounteney, I., Nightingale, P.D., Pereira, M.G., Olszewska, J., Pickard, A., Rees, A.P., Spears, B., Stinchcombe, M., White, D., Williams, P., Worrall, F., Evans, C., 2021. Landscape controls on riverine export of dissolved organic carbon from Great Britain. Biogeochemistry 2. https://doi.org/10.1007/s10533-021-00762-2

---

## Author Comment (AC5)

*Reviewer comment in italics*

Author response in plain text

**RC2 Comment**

*The manuscript of Pickard et al. focuses on the fluvial carbon export from disturbed and restored wetlands. As this carbon flux is a substantial factor in the carbon balance, they monitored dissolved organic and inorganic carbon as well as CO2 and CH4 in peatland draining streams over a two year period. The manuscript is well written and the generated data set is worth to be published. However, the presented study shows some major limitations, which has been addressed before in a pre-review and were not considered in a minor manuscript revision by the authors beforehand. Major concerns still are: 1) unclear way of calculation of carbon fluxes on all six monitoring sites, when only three were equipped with pressure transducer for water level recording; 2) reliability and clarity of the DIC and CO2 data 3) lack of data discussion and scientific hypotheses. These major points will be further addressed below. Especially point 2) raises my concerns and needs to be edited properly.*

Thank you for your comments – these have helped us to improve the manuscript.

*1) The description of p8 L151-156 is still not entirely clear to me. Do you conducted discharge measuring by dilution experiments on all six sites and correlated it to the three stage measurements? This will give you an error in the flux and FWM concentration data, which should be addressed somewhere.*

Yes, we conducted dilution experiments at all six sites, and produced correlations to the three sub-catchments with monitored water level. To show the process more clearly in line with the reviewer's comment, we have added the stage dilutions and resultant correlations to a supplementary information file. We also add further text to the discussion on the uncertainty that can be introduced through this method, although we are of the view that the stage correlations between sites are robust given that they produced fits of $r^2 > 0.84$ for all sub catchments.

*2) I can just repeat former questions and comments. If I understand correctly, DIC has been measured along with the DOC concentrations on filtered samples, which were stored up to 4 weeks! During this handling and storage, a lot happens to DIC, which is in equilibrium with CO2 in the atmosphere. Changes in pH, outgassing of CO2 and production in a non-sterile sample is most likely. I do not think that this kind of data meets scientific quality standards. Coming to measured CO2 concentrations, a dependency of CO2 speciation on pH in water samples is completely neglected. You state that you measured pH at each sampling. Why don't you use this information? Even more unclear to me is why you separate these two parameters, as normally CO2 (as calculated by Henrys Law) in solution is a major part of DIC under low pH (as this is probably the case in these catchments). Having pH and CO2 concentrations you can also calculate the entire DIC in the water sample. This might be more correct than the DIC measurements or could be used for validation. Additionally, in the results part these two measured parameters - CO2 and DIC - were summed up in the carbon export calculations. This is simply not correct as you double the CO2 contribution then. In the end, I don't see any sense in comparing CO2 concentrations at different sites when the pH is not considered.*

Carbon dioxide determination following the headspace method is a routine technique and it has recently been proven that for samples with low pH (<7.5, as is the case for headwaters draining peatland catchments) providing no correction for pH/alkalinity produces errors < 5% (Koschorreck et al., 2021). We have added this reference, and the associated quoted uncertainty, to the manuscript. It is in high pH samples where this issue becomes significant. We do not have alkalinity or pH data to

make this correction in any case. We have removed reference to pH sampling from the manuscript, as whilst these data were collected during sampling they are not available to us for use in the interpretation.

In regard to the separate analysis of DIC and $CO_2$, our rationale remains that we think it is of interest to measure these separately as dissolved $CO_2$ data are important for the potential evasion of carbon directly from headwater streams to the atmosphere, with clear implications for GHG budgets (and our data suggest that $CO_2$ is outgassed quickly, as indicated by the inverse relationship with catchment area drained – an important finding of the study). However, we take the reviewer's comments on board with regard to $CO_2$ being a part of the total DIC budget, and have now decided to present $CO_2$ data only. This hopefully also addresses the data quality concerns raised by the reviewer regarding the DIC data.

*3) There are some shortcomings in the scientific significance. The study presents a good data set, but is mainly descriptive and clear conclusions or benefits from the study are not well stressed. No hypotheses raised.*

We do not pose hypotheses, but we have introduced research questions at the end of the introduction which we aim to answer in the discussion. We believe that this is equally valid approach to hypothesis testing.

In terms of the scientific significance, we think that this data set clearly show the dramatic effect that drainage still has on aquatic peatland carbon losses even 50 years after ploughing of an undisturbed site. Restoration as a land use intervention inevitably leads to disturbance and a high degree of catchment spatial variability in terms of water and carbon flow, and our data show this to be the case. Noisy biogeochemical signals suggest that peatlands needs time to reset and reach a new equilibrium. We have added further emphasis on these points to the discussion to make this message clearer. We add other information on the scientific significance of the study from a carbon speciation perspective in the comment below.

*The relevance of different carbon species is not explained in the Introduction. Therefore, the research question how different carbon species vary and why it is important to measure them is not introduced. Moreover, the importance of DOC is highlighted before.*

One of the key outputs of this study is the carbon speciation data provided. We have added further information on the importance of each species to the introduction, namely that whilst DOC is typically the largest export term, POC can indicate erosional increases that can often be traced back to different land uses (i.e. drained peatland sites might be expected to have higher POC concentrations, and in some severely drained peatlands this can become the dominant C species contributing to total fluvial carbon losses ), and dissolved $CO_2$ and $CH_4$ have direct relevance for the GHG budgets of the streams themselves, as these gases are quickly evaded from solution to the atmosphere. Indeed, the consistent finding that dissolved gaseous GHG concentrations were higher in the smaller catchments is one of the interesting findings of this study – and to our knowledge, the first data of this kind collected in the Flow Country's headwater streams. In the revised manuscript we add previously unused nitrous oxide data to provide the full suite of dissolved GHGs. We hope that the additional text make the message about the importance of dissolved GHGs evading from peatland headwaters clearer, as this is a key finding.

*The discussion needs improvement and mainly cites literature from the UK. Maybe it would be helpful to additionally compare the study to international studies on rather natural sites, where more literature can be found? Another helpful publication might be: Swenson et al. 2019: Carbon balance*

*of a restored and cutover raised bog: implications for restoration and comparison to global trends, Biogeosciences 16 (3), p 713–731 DOI: 10.5194/bg-16-713-2019*

As we stated in response to reviewer 1, we have added a number of further references to the discussion to widen the scope beyond UK based references (e.g. Haapalehto et al., 2014; Menberu et al., 2017; Strack et al., 2015). Thank you also for the further paper suggestion – this has also been added.

*At last I would like to read some statements why it is reasonable to compare different carbon concentrations? As you cannot draw conclusions on the carbon balance or losses from the peatland it is probably because of water quality issues? What conclusions can be drawn from it?*

As we state above: "whilst DOC is typically the largest export term, POC can indicate erosional increases that can often be traced back to different land uses (i.e. drained peatland sites might be expected to have higher POC concentrations, and in some severely drained peatlands this can become the dominant C species contributing to total fluvial carbon losses ), and dissolved $CO_2$ and $CH_4$ have direct relevance for the GHG budgets of the streams themselves, as these gases are quickly evaded from solution to the atmosphere. Indeed, the consistent finding that dissolved gaseous GHG concentrations were higher in the smaller catchments is one of the interesting and novel findings of this study".

*Some specific comments: Fig 2. Tab. 2 and Fig 3 show all more or less the same data. Maybe you can reduce redundancy. The same goes for Table 5 and Figure 6.*

We disagree that Figures 2 and 3 show the same data as the boxplots give no indication of the temporal variation between the carbon species, and we think this is useful for the paper's readers to see. We have removed Figure 6 as it does show the same information as Table 5.

*P7 L 135-136: I am confused by the phrase "…affected by artificial drainage alone,..." I am no native speaker. Does this mean that that the non-drained catchments formerly has been drained and are additionally affected by other disturbances? Maybe you can clarify/rephrase this.*

It is meant to indicate that some sites have been artificially drained, whereas other sites have been artificially drained and afforested. We have amended the text to clarify this.

*P12, L214ff: As CO2 concentrations are greatly dependent on pH this should be considered here. Moreover, it would be nice to state if the water is supersaturated and outgassing prevails?*

As stated in another comment, we have now removed all reference to pH from the manuscript as we do not have the data available for interpretation.

*P 13 L233: link to "Figure 5" seems to be wrong*

This has been corrected.

*P 19 L302ff: "… with the second most important export component DIC followed by CO2" This makes no sense.*

As mentioned, we have now removed reference to DIC from the paper.

*P23 L414: see comment above*

See above.

**References**

Haapalehto, T., Kotiaho, J.S., Matilainen, R., Tahvanainen, T., 2014. The effects of long-term drainage and subsequent restoration on water table level and pore water chemistry in boreal peatlands. J. Hydrol. https://doi.org/10.1016/j.jhydrol.2014.09.013

Koschorreck, M., Prairie, Y. T., Kim, J., and Marcé, R.: Technical note: CO2 is not like CH4 – limits of and corrections to the headspace method to analyse pCO2 in fresh water, Biogeosciences, 18, 1619–1627, https://doi.org/10.5194/bg-18-1619-2021, 2021.

Menberu, M.W., Marttila, H., Tahvanainen, T., Kotiaho, J.S., Hokkanen, R., Kløve, B., Ronkanen, A.K., 2017. Changes in Pore Water Quality After Peatland Restoration: Assessment of a Large-Scale, Replicated Before-After-Control-Impact Study in Finland. Water Resour. Res. https://doi.org/10.1002/2017WR020630

Strack, M., Zuback, Y., McCarter, C., Price, J., 2015. Changes in dissolved organic carbon quality in soils and discharge 10 years after peatland restoration. J. Hydrol. https://doi.org/10.1016/j.jhydrol.2015.04.061